An application of topological data analysis in predicting sumoylation sites

Lin Xiaoxi lxx0217@mail.dlut.edu.cn
Gao Yaru
Lei Fengchun
School of Mathematical Sciences, Dalian University of Technology , Dalian , Liaoning , China
Chen Jun
Electronic publication date: 2023 Oct 12
Publication date: 2023
Volume: 11
Electronic Location ID: e16204
Received 2023 Apr 9; Accepted 2023 Sep 8
Copyright: ©2023 Lin et al.
Copyright year: 2023
Copyright holder: Lin et al.
License: This is an open access article distributed under the terms of the Creative Commons Attribution License, which permits unrestricted use, distribution, reproduction and adaptation in any medium and for any purpose provided that it is properly attributed. For attribution, the original author(s), title, publication source (PeerJ) and either DOI or URL of the article must be cited.
License URL: https://creativecommons.org/licenses/by/4.0/

Keywords: Topological data analysis, Sumoylation, Persistent homology, Feature extraction

Funding: NSFC and State Key Laboratory of Structural Analysis, Optimization and CAE Software for Industrial Equipment No.12071051 This work was supported by a grant (No.12071051) of the NSFC and the State Key Laboratory of Structural Analysis, Optimization and CAE Software for Industrial Equipment. There was no additional external funding received for this study. The funders had no role in study design, data collection and analysis, decision to publish, or preparation of the manuscript.

==============================
Sumoylation is a reversible post-translational modification that regulates certain significant biochemical functions in proteins. The protein alterations caused by sumoylation are associated with the incidence of some human diseases. Therefore, identifying the sites of sumoylation in proteins may provide a direction for mechanistic research and drug development. Here, we propose a new computational approach for identifying sumoylation sites using an encoding method based on topological data analysis. The features of our model captured the key physical and biological properties of proteins at multiple scales. In a 10-fold cross validation, the outcomes of our model showed 96.45% of sensitivity (Sn), 94.65% of accuracy (Acc), 0.8946 of Matthew’s correlation coefficient (MCC), and 0.99 of area under curve (AUC). The proposed predictor with only topological features achieves the best MCC and AUC in comparison to the other released methods. Our results suggest that topological information is an additional parameter that can assist in the prediction of sumoylation sites and provide a novel perspective for further research in protein sumoylation.

Introduction

Proteins typically require varying degrees of chemical modifications to perform functions after translation; these are called post-translational modifications (PTMs) (Zhao et al., 2014; Beauclair et al., 2015; Xu et al., 2016). PTMs increase the complexity of protein function in multiple ways, such as with the covalent addition of biochemical functional groups or proteins, the proteolytic cleavage of subunits, or the degradation of the whole protein (Chang et al., 2018). Sumoylation, an essential PTM, is done by the small ubiquitin-related modifiers (SUMOs). An increasing number of sumoylated proteins have been found in various eukaryotic cells in the past 20 years. Sumoylation is a reversible multi-step enzymatic reaction. To start this modification, the SUMO precursor is first cleaved into its mature form with the help of a family of SENP enzymes known as sentrin/SUMO-specific proteases. Based on the energy provided by the hydrolysis of ATP, mature SUMO forms a thioester bond with SUMO activating enzyme (E1). Then, activated SUMO is passed from E1 to SUMO conjugating enzyme (E2) to form a SUMO-E2 intermediate. Under the catalytic action of E3 ligase, the terminal glycine of SUMO is covalently linked to the free ϵ-amino group of a lysine in the substrate protein to be modified. SUMO can be deconjugated from the substrate protein with the help of SENPs, and then re-enters a new round of SUMO cycle. The schematic diagram of sumoylation and de-sumoylation is shown in Fig. 1. Various biological functions of protein are associated with sumoylation, such as nucleocytoplasmic transport, subcellular location, transcription, signal transduction, and the antagonism of ubiquitination (Seeler & Dejean, 2003; Hay, 2005; Kroetz, 2005). Sumoylation has a close connection with the development of human diseases, including congenital heart defeats (Wang et al., 2011), diabetes (Zhao, 2007), cancers (Seeler et al., 2007), and neurodegenerative diseases (Lee et al., 2013). Hence, the identification of sumoylation sites carries important implications for research on diseases and biomechanisms (Xu et al., 2016; Chang et al., 2018).

Figure 1 The SUMO cycle.

Several biochemical approaches have been introduced using purification strategies to identify sumoylation sites by employing epitope-tagged SUMOs or SUMO antibodies (Hendriks & Vertegaal, 2016; Hendriks et al., 2018). The problems with these approaches are that they are laborious and time consuming. Computational approaches are expected to provide economical, efficient, and accurate alternative to sumoylation experiments. In general, computational approaches can be clustered into sequence-based and structure-based groups. Structure-based methods incorporate learning algorithms with sequence features derived from BLOSUM62 (Xu et al., 2016; Zhu et al., 2022), pseudo amino acid composition (PseAAC) (Jia et al., 2016), position relative incidence matrix (PRIM) (Khan et al., 2021), etc. Structure-based methods integrate learning algorithms with physical features retrieved from local geometric information, such as half-sphere exposure (HSE) (Sharma et al., 2019), backbone torsion angles, accessible surface area (ASA) (Lopez et al., 2020), and contact number (CN) (Dehzangi et al., 2018).

Although the structure-based methods have assisted in the identification of sumoylation sites, they are often riddled with too much structural detail to be put into practice. Topological data analysis (TDA) offers a different strategy to characterize protein structures. Using a simplified topological model that is independent on metrics and coordinates, TDA can capture both local and global structural information of proteins. Persistent homology (PH), one of the key components of TDA, is a tool for data simplification and dimension reduction. The geometric information of an underlying space may be characterized by the persistence time of topological invariants through a process of filtration. Recently, PH-based methods have delivered some of the best results in computational biology, including the analysis and prediction of mutation-induced protein stability (Cang & Wei, 2017); protein thermal fluctuation and B-factor (Bramer & Wei, 2020); RNA data (Xia, Liu & Wee, 2023); RNA flexibility (Pun, Yong & Xia, 2020); protein secondary structure (Hassanpour, Izadkhah & Isazadeh, 2021); protein binding affinity (Nguyen et al., 2019); and chromosome packing, flexibility, and dynamics (Gong et al., 2022). These research findings indicate that TDA can effectively characterize biomolecular data and reflect biological and chemical properties, as well as others.

Our goal was to explore the utility and interpretability of the features constructed from TDA for identifying sumoylation sites. We constructed features that were able to describe the unique and meaningful properties of protein fragments, ranging from a local atom arrangement to its global architecture. In a 10-fold cross validation, the outcomes of our predictor had 92.78% of specificity (Sp), 96.45% of Sn, 94.65% of Acc, 0.8946 of MCC, and 0.99 of AUC, respectively. Our results suggest that topological information as an additional parameter may help to predict sumoylation sites. We further confirmed that sumoylation sites are closely related to the structure of proteins (Mann & Jensen, 2003; Sharma et al., 2019) from a topological standpoint.

Materials and methods

The following procedures were used to build a topology-based predictor that could effectively discriminate between sumoylation and non-sumoylation sites: (i) retrieve and preprocess datasets to train and test models; (ii) characterize each data sample with meaningful and distinguishable features constructed from TDA; (iii) train predictors based on machine learning algorithms; and (iv) evaluate the performance of predictors. The flowchart of our proposed methodology is shown in Fig. 2.

Figure 2 Flowchart of the proposed methodology.

Dataset description

In this study, we considered two sumoylation site datasets. One was retrieved from GPS-SUMO (Zhao et al., 2014) and consisted of 510 proteins with 912 annotated sumoylation sites; this dataset was named dataset1. This dataset was designed to explore the utility and interpretability of the features constructed from TDA for identifying sumoylation sites. The other dataset, named dataset2, was obtained from iSumok-PseAAC (Khan et al., 2021) with 4,987 annotated sumoylation sites from 1,311 proteins. The applicability of our proposed model was validated using dataset2. As not all complete atomic coordinates of every protein were accessible, 471 proteins (Data S1) and 1,288 proteins (Data S2) with their complete coordinates and sequence information, were retained in dataset1 and dataset2, respectively. Relevant information is available on the AlphaFold database (Varadi et al., 2022) and the UniProt database (Wang et al., 2021).

The preprocessing procedures for each dataset were as follows. First, each lysine K from a protein was characterized by a peptide, P, whose length was selected based on the work of Khan et al. (2021). P was composed of 20 upstream and downstream residues, respectively, with K as the center. Missing residues were added dummy code X. The peptide, P, was considered to be a positive sample if its center, K, was experimentally annotated as a sumoylation site; otherwise, it was considered to be a negative sample. Further, we mixed the positive and negative samples and computed the pairwise sequence identity to avoid the bias of homology. If the sequence identity between two given samples was more than 40%, only one of them was retained while the other was ignored. Ultimately, the samples that were not redundant were divided into a positive subset or a negative subset according to the category to which they belonged. After going through these steps, we obtained 775 positive samples (Data S3) and 17,807 negative samples (Data S4) from dataset1, and 4,493 positive samples (Data S5) and 24,456 negative samples (Data S6) from dataset2.

Preparations

The features constructed from TDA for identifying sumoylation sites were obtained as follows. First, the original protein data was represented in an all-atom model. Then, simplicial complexes were constructed according to the represented data. Afterwards, the PH analysis was conducted to reveal the topological information of the proteins. Finally, features were extracted from persistence barcodes (PBs), which could visualize the results of the PH analysis. Additional information on TDA and PH can be found in Edelsbrunner, Letscher & Zomorodian (2000), Zomorodian & Carlsson (2004), Munkres (2018), and Wang, Cang & Wei (2020).

Representation of protein data

We used an all-atom model when dealing with the protein fragments. An all-atom model includes various types of atoms, such as O, C, N, S, and P, which are all of equal importance. Note that the hydrogen atoms were ignored during the PH analysis, as they created redundant barcodes that did not contribute much to feature construction. Figure 3 shows two protein fragments of O00429 (UniProt Entry) and their corresponding all-atom models. The central lysine of each fragment is K532 (sumoylation site) and K92 (non-sumoylation site), respectively.

Figure 3 All-atom models of fragments of O00429.

Simplicial complexes

Figure 3 illustrates that the represented protein fragments are essentially the point cloud data of dimension 3. The Vietoris-Rips (VR) complex and Alpha complex were considered to characterize the point cloud data in this work. The following definitions also can be seen in Pun, Lee & Xia (2022).

Let X be a finite point set in Euclidean space ℝn. The VR complex of X with parameter ϵ is the set of all σ⊆X, such that any pairwise distance of its points is at most 2ϵ.

To introduce the Alpha complex, we need some related concepts. Given a good cover U of X, i.e., X⊆∪i∈IUi. The nerve of U is defined as: (1) NU=J⊆I∣∩j∈JUj≠0̸∪0̸.

A closed ball in ℝn with center, x, and radius, δ, is denoted as B(x, δ). The union of closed balls with center points in X forms a cover of X, and the corresponding nerve creates a simplicial complex named the Čech complex, (2) CX,δ=σ⊆X∣∩x∈σBx,δ≠0̸.

Given a point, x ∈ X, the Voronoi cell of x is defined as: (3) Vx=y∈Rn∣|y−x|≤|y−x′|,∀x′∈X.

The Voronoi diagram is the collection of all Voronoi cells. The dual graph of the Voronoi diagram forms a simplicial complex called the Delaunay complex. Let R(x, δ) be the intersection of the Voronoi cell, Vx, with the ball, B(x, δ), that is, R(x, δ) = Vx∩B(x, δ). The Alpha complex of X is defined as the nerve of cover ∪x∈XR(x, δ), i.e.: (4) AX,δ=σ⊆X∣∩x∈σRx,δ≠0̸.

Intuitively, the Alpha complex is a subcomplex of the Delaunay complex.

To illustrate the theory of simplicial complexes, we consider the example showing a set of points, S, of dimension 2 (Fig. 4A). With the increasing radius, the simplices contained in the two complexes have distinct differences. The VR complex is entirely determined by its 1-simplices, that is, if all the 1-faces of a simplex are in the VR complex, then so is the simplex. However, the Alpha complex is only suitable for subjects in Euclidean spaces, and its construction is more complicated. Based on the same dataset, diverse simplicial complexes can be constructed according to different rules. Therefore, it may be of value to combine these complexes for the PH analysis, as they may reveal different information about the same protein data.

Figure 4 The comparison of simplicial complexes.

(A) Point set S in ℝ2. (B) The simplicial complex is constructed from S by giving each ball a small radius. The VR complex is the same as the Alpha complex, which only has six 0-simplices. (C and D) The VR complex and Alpha complex constructed from S by giving each point a large radius, respectively. Each convex region in (D) divided by the dotted lines contains a point in S, and represents the Voronoi cell of that point. (Black, red, green, and blue colors denote the 0-, 1-, 2-, and 3-simplices, respectively.)

(Element specific) persistent homology analysis

Typically in PH, a nested sequence of subcomplexes is constructed based on a filtration parameter. After analyzing the homology of each subcomplex, the topological information is characterized by different persistence time of homological generators with respect to the filtration parameter.

Given a simplicial complex, K, the filtration of K is a nested sequence of subcomplexes of K, i.e.: (5) 0̸=K0⊆K1⊆⋯⊆Kn=K.

The p-persistent k-th homology group at filtration time i can be represented as: (6) Hki,p=Zki/Bki+p∩Zki,

where Zki is the k-th cycle group of Ki and Bki+p is the k-th boundary group of Ki+p. The rank of Hki,p is called the p-persistent k-th Betti number of Ki, denoted as βki,p.

In the PH results, each topological generator is characterized by a pair of values that record when it appears and dies, named birth time (BT) and death time (DT), respectively. The PH results can be visualized as PBs using the endpoints of a bar to represent the BT and DT of each generator, respectively. Figure 5 shows the PBs of dimension 0, 1, and 2 of K532 and K92 from the filtration of the VR complex, respectively. With the increasing filtration values, atoms are pairwise connected according to different interactions of the protein. These connections induce the death of generators of dimension 0. More connections indicate a greater occurence of simplices of higher dimensions, which are related to the 1- and 2-bars. There is a clear difference between the 0-bars in [1.25, 1.5] Å of Figs. 5A and 5B that reflects the different interaction patterns of these fragments. Moreover, more types of 1- and 2-bars can be found in the lower right panels, indicating that K92 has a more complicated spatial structure, which is consistent with Fig. 3.

Figure 5 Persistence barcodes.

(A) The 0-, 1-, and 2-bars of K532. (B) The 0-, 1-, and 2-bars of K92.

We adopted the element specific persistent homology (ESPH) analysis to reveal additional information about the biochemical and physical properties of the proteins. ESPH analyzes biomolecular data by specifying one or more types of elements, followed by the PH analysis (Meng et al., 2020). Distinguishing element types enables us to reduce biomolecular complexity and retain critical properties of the studied data (Cang & Wei, 2018). In this work, element C and N were considered for ESPH, and the software package GUDHI (Project, 2021) was used for the (ES) PH analysis.

Feature extraction

PBs cannot be used as direct input to learning algorithms, therefore, it must be converted into feature vectors to train a predictor. Additionally, the features extracted from PBs should reflect meaningful and distinguishable information. In this work, two common vectorization methods were considered.

The binning approach (BA), which discretizes the filtration domain into various sizes of bins, was used. Bins of dimension 1, i.e., [xi, xi+1], i =0 , 1, …, n − 1, with x0 = 0 and xn = rf (rf denotes the ending value of filtration), were used to vectorize PBs. The number of bars of a given dimension, whose death time are within a given bin, serves as an entry of a feature vector. This method allows for the detection of different protein interactions with a wide range of scales, such as a hydrogen bond, van der Waals, and hydrophilic and hydrophobic reactions (Cang & Wei, 2017).

Barcode statistics (BS) were also used. This method summarizes the statistics of barcodes. The maximum, minimum, mean, summation, and standard deviation of BTs, DTs, and bar lengths (BLs) were considered here. These statistics were used to vectorize 0-, 1-, and 2-bars from the filtration of the VR or Alpha complexes.

Features based on topology

For a peptide sample, P, the features of P were divided into four categories according to different vectorization methods and filtrations as follows:

- TF1: features based on the 0-, 1-, and 2-bars of P from the filtration of the VR complex.

- TF2: features based on the 1- and 2-bars of P from the filtration of the Alpha complex.

- TF3: features based on each residue of P (excluding the central lysine K) from the filtration of the VR complex.

- TF4: features based on the local region of P from the filtration of the VR complex.

In TF1, 1-bins (unit: Å) were used to vectorize 0-, 1-, and 2-bars. The endpoints of the bars were taken from two consecutive values in the specific list for different dimensions. These lists were set to [1.2, 1.3, 1.4, 1.5, 1.6, 2.0], [1.5, 2.7, 3.5, 4.5, 5, 6.7], and [2.4, 2.9, 5.5, 6.7], respectively. For example, (1.2, 1.3) Å was used as a bin to vectorize 0-bars of P. Moreover, TF1 included the second and third longest bar lengths of dimension 0, the sum and mean of bar lengths of dimension 0, the onset value of the longest 1-bar, and the barcode statistics of 1- and 2-bars. Therefore, TF1 contained 48 features. The barcode statistics of 1- and 2-bars from the filtration of the Alpha complex resulted in 30 features in TF2.

In TF3, for each residue of P, the (1.25, 1.5] Å and (1.5, 1.75] Å bins were used. TF3 also contained the number of 0-bars, and the summation of bar lengths of dimension 0, 1, and 2. Note that each feature of the dummy code X was set to 0. Hence, the number of features in TF3 was equal to 240.

In TF4, the local region of P refered to the peptide fragment from the 2-th upside residue to the 2-th downside residue, according to the central lysine K. For the PH analysis, 1-bins were taken from the (1.2, 1.6) Å split by the fixed bin size 0.1 Å, and the barcode statistics were used to vectorize the 1-bars. For the ESPH analysis of element type C, 1-bins were taken from the (1.5, 3) Å split by the fixed bin size 0.5 Å, and the barcode statistics were used to vectorize the 1-bars; for the ESPH analysis of element type N, the number of 0-bars whose death times were less than 10 Å was considered. TF4 contained 38 features. All of these four categories gave rise to a total of 356 features for P.

Model training and validation

There was an imbalance of the sample size of positive and negative classes for each dataset, which may affect the learning process. To address this issue, an undersampling method named NearMiss (Lopez et al., 2020) was used. The “imbalanced-learn” package (Lemaître, Nogueira & Aridas, 2017) of Python was employed to balance each dataset. After undersampling, 775 and 4,493 negative samples were selected from dataset1 and dataset2, (Data S7 and S8), respectively.

The features constructed from TDA were fed into various binary classifiers, including the gradient boosting classifier (GBC), random forest classifier (RFC), and support vector classifier (SVC). GBC uses a boosting algorithm to make up for the shortcomings of the original model by building a weak learner at each step of the iteration. RF is an ensemble learning method based on the bagging algorithm, which independently integrates different decision trees during training. SVC is a classifier based on the support vector machine algorithm whose decision boundary is the maximum-margin hyperplane solved for the learning samples. All of these classifiers were implemented here with the “scikit-learn” package (Pedregosa et al., 2011) of Python. The “n_estimators” parameter was set to 370 and 500 for RFC and GBC, respectively. Moreover, for RFC, the “oob_score” parameter was “True”, and the “max_features” parameter was chosen as “sqrt”; for SVC, the “gamma” parameter was set to “auto”, and the “probability” parameter was “True”. All other parameters used their default values.

The K-fold cross validation and independent set test were adopted to evaluate the model performance here. The K-fold cross validation splits the original dataset into K disjoint subsets. Each subset is selected in turns for testing, while the rest of parts are used for training. Note that each data participates in training in the K-fold cross validation. However, the independent set test divides the original dataset into training and testing subsets at a given dividing ratio, where samples in the testing subset are not participants in model training.

Evaluation metrics

To evaluate the performance of predictors from different perspectives, we adopted multiple evaluation metrics as follows: (7) Sp=TNTN+FP,

(8) Sn=TPTP+FN,

(9) Acc=TP+TNTP+TN+FP+FN,

(10) MCC=TP×TN−FP×FNTP+FPTP+FNTN+FPTN+FN,

where “T” and “F” denote the true and false cases of prediction, “P” and “N” denote the positive and negative classes, respectively. Specifically, TP (true positive) counts the correctly predicted sumoylation sites, and TN (true negative), FP (false positive), and FN (false negative) are defined similarly.

The performance of predictor was also measured using the area under the receiver operating characteristic (ROC) curve. Different pairs of true positive rate (TPR, also known as Sn) and false positive rate (FPR, defined as follows) can be obtained by adjusting the classification threshold of a given classifier, which are the data points of ROC. The AUC of ROC refers to the probability that a classifier outputs a higher probability for the given positive sample being positive than for the given negative sample being positive, when randomly given a positive and a negative sample. It reflects the sorting ability of a classifier. A higher AUC implies a better classifier. (11) FPR=FPFP+TN.

In addition to evaluating model performance, we sought to determine the topological features that contributed to the prediction of sumoylation sites. The F-score value (Xu et al., 2016) measures the ability of features to distinguish among two classes. The F-score value of the ith feature is defined as: (12) Fi=φi¯+−φi¯2+φi¯−−φi¯21n+−1 ∑k=1n+φk,i+−φi¯+2+1n−−1 ∑k=1n−φk,i−−φi¯−2i=1,2,…,Ω,

where φk,i+ and φk,i− are the ith feature value of the k-th positive sample and k-th negative sample, respectively. φi¯+ is the mean of all φk,i+, i.e., φi¯+=1n+∑k=1n+φk,i+.φi¯− can be obtained similarly. φi¯ denotes the mean of the ith feature values of all samples, that is, φi¯=1n++n−∑k=1n+φk,i+∑l=1n−φl,i. The higher F-score value means the greater contribution to the classification. The codes of our work are available on the GitHub repository using the link: https://github.com/Xiaoxi-Lin/SUMO_TOP.git.

Results

We first explored the utility of the features constructed from TDA on dataset1. The F-score values of all 356 features were calculated and sorted from high to low (Table S1). We then added features one at a time to generate different feature sets according to the sorted F-score values. For each feature set, we trained and evaluated the predictor by employing GBC and 10-fold cross validation, respectively. Figure 6 shows the MCC values based on different feature sets, where MCC gets the maximal value 0.8946 when the top 352 features are used. Finally, these 352 features were selected as optimal features for our subsequent analysis of the interpretability of features.

Figure 6 The index feature score curve.

The proposed GBC predictor with the features constructed from TDA was called SUMO_TOP in this work. In a 10-fold cross validation on dataset1, the outcomes of our predictor were 92.78%, 96.45%, 94.65%, 0.8946, and 0.99 of Sp, Sn, Acc, MCC, and AUC, respectively. Compared with existing methods, including SUMO_sp2.0-H (Ren et al., 2009), GPS_SUMO-L (Zhao et al., 2014), JASSA (Beauclair et al., 2015), SUMO_LDA (Xu et al., 2016), pSUMO_CD (Jia et al., 2016), HseSUMO (Sharma et al., 2019), and iSumok-PseAAC (Khan et al., 2021), SUMO_TOP delivers a comparable performance. The results of these predictors are shown in Table 1. The performance metrics, Sp and Sn, are dependent on each other (Chou, 1993). Additionally, a higher Sp (Sn) but lower Sn (Sp) may result in a higher Acc. Therefore, a meaningful comparison should consider the rate of their combination, that is, the score of MCC (Jia et al., 2016). SUMO_TOP is shown to have the maximal MCC and AUC values (see Fig. 7A), which reflects its usefulness in practical applications. It also indicates that the proposed predictor can effectively classify sumoylation and non-sumoylation sites.

Table 1 The evaluation indicators of SUMO_TOP and other existing methods.

The best results are indicated in bold.

Methods	Sp (%)	Sn (%)	Acc (%)	MCC	AUC	
SUMO_TOP	92.78	96.45	94.65	0.8946	0.99	
iSumok-PseAAC	94.51	94.24	94.79	0.8903	0.96	
pSUMO_CD	99.21	82.01	97.88	0.8460	–	
GPS_SUMO-L	66.80	81.00	73.90	0.7780	0.87	
HseSUMO	87.2	90.4	88.8	0.776	–	
SUMO_LDA	84.51	98.71	86.92	0.6845	–	
SUMO_sp2.0-H	60.8	87.3	74.0	0.498	0.73	
JASSA	65.4	80.8	77.3	0.467	0.73	

Figure 7 ROC curves and AUC values.

(A) The ROC curves of SUMO_TOP with 10-fold cross validation on dataset1. (B) The AUC values of SUMO_TOP with 50 independent set tests on dataset1. (C) The ROC curves of predictors with the independent set test on dataset2. (D) The AUC values of predictors with 5- and 10-fold cross validations on dataset2.

In contrast to other methods with hybrid types of sequence or structural features, the present work only used the features of peptides constructed from TDA. This information characterizes sumoylation sites from a topological view. Results obtained on dataset1 preliminarily verify the utility of the features constructed from TDA for predicting sumoylation sites. The features constructed from TDA may be further employed to improve the performance of other methods listed in Table 1. The topological information may be combined with other established features, such as AAindex, PseAAC, and HSE, to capture the structural and biochemical properties of sumoyaltion sites from a more holistic perspective. A more efficient and accurate model may be constructed for identifying sumoylation sites using these features.

We also adopted different binary classifiers and validation strategies to varify our results. Table 2 records the means and standard deviations of the evaluation indicators of SUMO_TOP obtained by 50 independent set tests with different test-sizes. The corresponding AUC values are shown in Fig. 7B. The results of other predictors in 5- and 10-fold cross validations are given in Table S2. These similar results reflect that the feature construction based on TDA is a stable and robust encoding method for predicting sumoylation sites.

Table 2 The results of 50 independent set tests of SUMO_TOP.

Test-size	Sp	Sn	Acc	MCC	
0.2	0.966 (±0.014)	0.916 (±0.024)	0.941 (±0.014)	0.883 (±0.026)	
0.25	0.965 (±0.012)	0.915 (±0.018)	0.940 (±0.010)	0.882 (±0.019)	
0.3	0.966 (±0.012)	0.914 (±0.017)	0.940 (±0.010)	0.881 (±0.020)	

To further explore the applicability of our proposed model, we applyed SUMO_TOP to dataset2. In this research, the 70:30 ratio was used for training (Data S9) and testing (Data S10), and measured the highest MCC value which is given in Table 3. SUMO_TOP also achieved comparable results. It further verifies that the utility of the features constructed from TDA for predicting sumoylation sites. Results of different predictors under the independent set test are shown in Table 4 and Fig. 7C. Table S3 records the performance indicators of other predictors on dataset2 in 5- and 10-cross validations, and Fig. 7D shows the related AUC values. Based on the results on the two datasets, we suggest that topological information as an additional parameter could assist in predicting sumoylation sites.

Table 3 Comparison of the independent set test.

The best results are indicated in bold.

Methods	Sp (%)	Sn (%)	Acc (%)	MCC	AUC	
Sumo_TOP	91.24	85.51	88.35	0.7685	0.95	
iSumok-PseAAC	89.29	88.16	88.60	0.7651	0.94	

Table 4 Independent set test of various predictors with topological features.

Predictors	Sp (%)	Sn (%)	Acc (%)	MCC	AUC	
SVC	91.77	77.43	84.53	0.6985	0.91	
RFC	89.52	80.22	84.83	0.7000	0.92	
GBC	91.24	85.51	88.35	0.7685	0.95	

Discussion

In the optimal feature set, each category had 47, 30, 237, and 38 features, respectively. The distribution of each category is shown in Fig. 8A. The large proportion of the third category is attributed to the largest number of features in TF3. Figure 8B shows the number of features selected as optimal features in each category, and the ratio is 0.98, 1, 0.99, and 1, respectively.

Figure 8 Distributions of features.

(A) The distribution shows the ratio of each category in the optimal feature set. (B) The number of features selected as optimal features in each category.

The features of TF1 revealed abstract pairwise atomic interaction patterns. Features in TF2 were capable of identifying the geometric information of protein fragments, such as voids and cycles. Features based on residues were able to capture local structural properties of amino acids (AAs), such as the pentagonal ring or hexagonal ring, and to further identify the types of AAs. Features in TF4 were related to the properties of the local region of protein fragments, such as charge and hydrophobicity (Cang & Wei, 2017). Almost all of the features belonging to TF1 and TF2 were selected into the optimal feature set, which illustrates that the structural features of proteins may be used to predict sumoylation sites (Sharma et al., 2019). The characteristics of AAs in the protein fragment were shown to be an important parameter with which to predict sumoylation sites (Khan et al., 2021) according to the large proportion of TF3. All of the features in TF4 were part of the optimal feature set, indicating that the features of the local region of peptide play a critical role in the recognition of sumoylation sites (Zhu et al., 2022).

Table 5 lists the top ten features of SUMO_TOP according to the reverse-sorted F-score values. Features 2, 4, 5, 6, 7, and 8 are derived from TF1. TF1 characterized the biomolecular structures of protein fragments, which were capable to capture important biological properties. It could also detect the secondary structure of protein fragments to some extent (Hassanpour, Izadkhah & Isazadeh, 2021; Pun, Lee & Xia, 2022), which implies that the secondary structure of protein may provide important information on the interactions of AAs along the protein sequence (Dehzangi et al., 2018). Features 1, 3, 9, and 10 belong to the fourth category; the first three are the results of the ESPH analysis of element C. It could effectively capture the hydrophobic reactions and changes in the local region of peptides (Cang & Wei, 2017). Hydrophobicity was also shown to be useful to the prediction of sumoylation sites (Chen et al., 2012) from a topological view.

Table 5 Features with the top ten F-score values.

Index#	Category#	Betti#	Complex	Features	
1	4	1	VR	mean of DTs of atom C	
2	1	2	VR	mean of BLs	
3	4	1	VR	mean of BTs of atom C	
4	1	2	VR	min of BLs	
5	1	2	VR	standard deviation of BTs	
6	1	1	VR	standard deviation of BTs	
7	1	2	VR	standard deviation of DTs	
8	1	2	VR	max of BLs	
9	4	1	VR	standard deviation of BTs of atom C	
10	4	1	VR	sum of BLs	

Conclusions

PH is a tool for exploring topological characteristics by studying the sequence of nested simplicial subcomplexes. The PH analysis in our work was capable of capturing the structural information of proteins from multiple angles and scales. Based on the understanding of protein interactions at different scales (Xia & Wei, 2015), we attempted to apply the features constructed from TDA to predict sumoylation sites. To our knowledge, it is the first time that TDA has been used to predict PTM sites.

The proposed tool was used to predict protein sumoylation sites. We retrieved two datasets, where each peptide sample was formulated into 356 features constructed from TDA. Our predictor shows comparable performance with other existing methods. It is worth noting that only features constructed from TDA were used in our predictor, instead of hybrid types of existing features. Moreover, our proposed model yields similar results under various validation strategies, which illustrates that the feature construction based on TDA is a stable and robust encoding method for predicting sumoylation sites. As a new application of TDA, our work suggests that topological information as an additional parameter could assist in the prediction of sumoylation sites. It further indicates that computational topology combined with machine learning might create a novel perspective for biomolecular study.

Our work examined the utility, efficiency, and interpretability of the features constructed from TDA for predicting sumoylation sites. As such, only topological information was employed. There are various strategies to improve our method. For instance, the features constructed from TDA may be combined with other established features, such as sequence and physical features, and a combination of these features may provide better results in the prediction of sumoylation sites.

Supplemental Information

Supplemental Information 1 Ranked F-score values of features constructed from TDA

Click here for additional data file.

Supplemental Information 2 Results of predictors with 5- and 10-fold cross validation on dataset1

Click here for additional data file.

Supplemental Information 3 Results of predictors with 5- and 10-fold cross validation on Dataset 2

Click here for additional data file.

Supplemental Information 4 Sumoylation sites with their corresponding uniprot entries of Dataset 1

Click here for additional data file.

Supplemental Information 5 Sumoylation sites and their corresponding peptides of Dataset 2

Click here for additional data file.

Supplemental Information 6 De-redundant positive samples of Dataset 1

Click here for additional data file.

Supplemental Information 7 De-redundant negative samples of Dataset 1

Click here for additional data file.

Supplemental Information 8 De-redundant positive samples of Dataset 2

Click here for additional data file.

Supplemental Information 9 De-redundant negative samples of Dataset 2

Click here for additional data file.

Supplemental Information 10 Negative samples of undersampled Dataset 1

Click here for additional data file.

Supplemental Information 11 Negative samples of undersampled Dataset 2

Click here for additional data file.

Supplemental Information 12 Training set of independent set test on Dataset 2

Click here for additional data file.

Supplemental Information 13 Testing set of independent set test on Dataset 2

Click here for additional data file.

We would like to thank the referees for their helpful comments and suggestions.

Additional Information and Declarations

Competing Interests

Author Contributions

Data Availability

The authors declare there are no competing interests.

Xiaoxi Lin conceived and designed the experiments, performed the experiments, analyzed the data, prepared figures and/or tables, authored or reviewed drafts of the article, and approved the final draft.

Yaru Gao conceived and designed the experiments, authored or reviewed drafts of the article, and approved the final draft.

Fengchun Lei conceived and designed the experiments, authored or reviewed drafts of the article, and approved the final draft.

The following information was supplied regarding data availability:

The data is available in the Supplemental Files.

The code is available at GitHub:

- https://github.com/Xiaoxi-Lin/SUMO_TOP.git

- Xiaoxi-Lin. (2023). Xiaoxi-Lin/SUMO_TOP: v1.0.0 (v1.0.0). Zenodo. https://doi.org/10.5281/zenodo.8035475

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
