# Peer review of "An application of topological data analysis in predicting sumoylation sites"

_PeerJ, doi:10.7717/peerj.16204_

## Round 0.1 · original submission · Major Revisions

Our reviewers are quite positive regarding your work, and have several important suggestions for its improvement. I suggest you pay especial attention to the comments regarding validation of the method in the prediction of other known sumoylation sites, as well as those regarding proper documentation of your methods files so that it can be immediately used by other workers upon download.

·

Basic reporting

In recent years, the impact of topological data analysis (TDA) in sciences and engineering has grown exponentially. The main tool of TDA, persistent homology (PH), helps bridge the gap between complex geometry and abstract topology through filtration. PH has been incredibly successful in handling intricately complex, high-dimensional, nonlinear, and multiscale data. However, it has some limitations, including its inability to handle heterogeneous information (i.e., different types of objects in the point cloud), its qualitative nature (e.g., a 5-member ring is counted the same as a 6-member ring), its lack of description of non-topological changes (i.e., homotopic shape evolution), its inability to cope with directed networks and digraphs, and its inability to characterize structured data (e.g., self-organizations in data). These limitations hinder the applications of PH in biomolecules. Element-specific persistent homology (ESPH) introduced by Cang et al can address some of these challenges. The present work further explores ESPH for an important biological problem.

Experimental design

Post-translational modification is an important process in biology. Small ubiquitin-related modifiers (SUMOs) are commonly involved and the related process is called
sumoylation. The authors developed ESPH to characterize predicting sumoylation sites. ESPH is capable of capturing the structural information of proteins from multiangles and at multiscales. The structural dataset has 766 sumoylation sites and 17769 non-sumoylation sites. ESPH is employed to extract 278 features from the data. Random forest model is used for machine learning classification and feature selection. An optimal set of 102 features was used the final prediction.

Validity of the findings

The TDA based method offers very good results compared with other methods. This is the first application of TDA to sumoylation analysis. The approach demonstrates the utility, efficiency and interpretability of ESPH for the prediction of sumoylation sites. This paper is well-written. I recommend it publication with a minor suggestion.

Additional comments

I would like to point out that before the development of ESPH, handcrafted PH features were also designed for protein classification by Cand et al (https://doi.org/10.1515/mlbmb-2015-0009). It is one of the first papers that combine machine learning and TDA.

Reviewer 2 ·

Basic reporting

no comment

Experimental design

no comment

Validity of the findings

no comment

Additional comments

Comments:
In this study, the authors constructed a topology-based predictor (SUMO_TOP), which is the first time that TDA used in predicting sumoylation sites. Although only topological information is employed in SUMO_TOP, its delivers a comparable performance to existing methods. However, several problem might be solved.

1. In the dataset description part, the author mentioned that to avoid homology bias, the peptide samples in the same positive or negative subsets have <40% pairwise sequence identity. But it is better to mix the positive and negative samples and perform the same operation.
2. Table 1 showed multiple evaluation metrics for various classifiers. It is required to show the standard deviation for comparison.
3. Please use statistical methods to prove the superiority of the proposed model.
4. The authors used ~1000 sumoylation sites. However, there are more than 10,000 reported sumoylation sites. Please use more the reported data to prove the universal applicability of the model.
5. The raw data and code needs to be uploaded to Github with a detailed description of usage.
6. The fonts in the images in the article need to be consistent, as required for PeerJ publication. Figure 4 is missing information on the horizontal and vertical axes. Figure 5 needs to be increased in resolution and kept at normal scale.
7. A diagram of the overall experimental design should be provided.
8. Figure 1 is described/explained in lines 29-34. But it is complex and the detailed description is required.
9. The column "Threshold" in Table 1 has no clear explanation. Please provide more details.
10. In line 182, the sentence "which is done by a screening procedure" is not cleary expressed. The authors should mention the name of the "screening procedure".
11. Figure 5 is not clearly visible. It should use a magnifying glass, highlighting the maximum MCC value.
12. Supplemental files include 510 proteins with 912 annotated sumoylation sites (Data S1), 766 positive samples (Data S2) and 17769 negative samples (Data S3). It’s missing a negative sample data after undersampling.

Reviewer 3 ·

Basic reporting

In this paper, the authors develop a Topological data analysis (TDA) based machine learning model for the prediction of sumoylation sites. More specifically, different from all previous models, the authors build simplicial complexes for sumoylation sites and perform the persistent homology analysis. The topological features are systematically generated from persistent barcodes, and further used as input for machine learning model, in particular, random forest model. It has been found that the prediction is highly accurate with with sensitivity, accuracy, Matthew's correlation as 91.27%, 94.26%, and 0.8877, respectively.
The paper is well written and the results are very promising. I would recommend the publication of the paper if the following questions were well addressed.
Major concerns,
1) In the introduction part, the authors give a general description of the application of the TDA in computational biology. There are some important areas that have not left out. In particular, persistent homology has been used in the analysis of RNA properties,
Pun, Chi Seng, Brandon Yung Sin Yong, and Kelin Xia. "Weighted-persistent-homology-based machine learning for RNA flexibility analysis." Plos one 15.8 (2020): e0237747.
Xia, Kelin, Xiang Liu, and JunJie Wee. "Persistent Homology for RNA Data Analysis." Homology Modeling: Methods and Protocols. New York, NY: Springer US, 2023. 211-229.
Further, persistent homology has been used in chromosome packing, flexibility and dynamics analysis,
Gong, Weikang, et al. "Persistent spectral simplicial complex-based machine learning for chromosomal structural analysis in cellular differentiation." Briefings in Bioinformatics 23.4 (2022): bbac168.
2) In table 1, the authors systematically compare with existing models. It is mentioned that they have undersample the original dataset to generate 766 positive and 766 negative samples. Are the other models in the comparison use the same training and test dataset?
Minor concerns,
1) The reference
Pun, C. S., Xia, K., and Lee, S. X. (2018). Persistent-homology-based machine learning and its 390 applications–a survey. arXiv preprint arXiv:1811.00252. DOI: 10.48550/arXiv.1811.00252
Should be updated to the new version,
Pun, Chi Seng, Si Xian Lee, and Kelin Xia. "Persistent-homology-based machine learning: a survey and a comparative study." Artificial Intelligence Review 55.7 (2022): 5169-5213.
2) In figure 3, it seems the generated Alpha complex should be the same as the VR complex. The 2-simplex in VR should also be in the Alpha complex (as the three balls overlap!) The authors are suggested to double check on it. Maybe consider reduce the radius of the balls, so that they are pair-wise overlap but do not share common region.
3) Figure 4, it seems Betti 2 is used in the later computational. The authors are suggested to add the results for Betti 2. Further, it is better to add the unit in the x-axis.
4) Table 2, what is the meaning for “Atom C with bin [2.5 3]”? If is the total number of bins, then the unit should not be Angstrom. Note that notation above is “Description (Angstrom)”. Better change it to “Features”, and do not need the unit, as some of the features many not be measured in terms of length.

Experimental design

The experimental design is proper and is in accordance with the general machine leaning procedure.

Validity of the findings

The findings are reasonable and convincing.

Additional comments

NA

Reviewer 4 ·

Basic reporting

Figure 1 needs to be improved. Its not suitable for publishing in terms of quality and resolution.

The is some ambiguity in the structure and flow of information in the article. The author discusses how data feature set is prepared before discussing how data itself is gathered. I think the sequence can be improved for better readability if it does not conflict with any given structure of the journal.

Experimental design

The author discusses the use of Random Forest. But details regarding the working of RF are not given. Further for reproducibility of results its important that exact parameters used for training of RF are provided.

Only one classification model is used by the author. It would be interesting to see results on other state of art classification models as well.

The authors have used the dataset compiled by Zhao et al. which is almost 9 years old. I am sure there must have been many additions to the dataset since then. The authors should extract an updated listing from Uniprot or some other resource.

Validity of the findings

The testing needs to be rigorous and comprehensive. Authors have provided results of 10 fold cross-validation only. In order to build a more convincing case for the readers the authors should performs more test including 5-fold cross validation, indepoendent set testing and jackknife testing.

A comparative analysis needs to performed with other existing models used for identification of Sumoylation sites.

Additional comments

The work is useful for research community. However the above given suggestions maybe incorporated before publication.

---

## Round 0.2 · Minor Revisions

You have addressed all the reviewers' comments and the reviewers are satisfied with the revision.

However, one of the Section Editors has identified that the manuscript needs editing for English. Their comments are as follows:

>The manuscript cannot be Accepted without further Editing. For example, here is a sample edit for the abstract:
>
>"Sumoylation, a reversible post-translational modification, regulates various biochemical functions. Some human diseases have been associatied with alterations in protein sumoylation. Hence, the identification of sumoylation sites in proteins provides insights for further mechanistic research in sumoylation and drug development. In this paper, we propose a new computational approach for predicting sumoylation sites through the features constructed from topological data analysis. The features capture important physical and biological properties of proteins at multiple scales. In a 10-fold cross validation, our predictor has improved sensitivity, accuracy, and ???, with a Matthew’s correlation coefficient equal to 96.45%, 94.65%, 0.8946, respectively. The proposed predictor with only topological features yields comparable results with other released methods. The data suggest that topological information as an additional parameter may assist the prediction of sumoylation sites and provide a novel perspective for further research in protein sumoylation."
>
>Also, in the sentence discussing a 10-fold cross validation, you mention only two features but 3 metrics, so this sentence need editing.
>
>Please ask the authors to edit the entire manuscript accordingly.
>
>Also, I was unable to understand how this application improves current sumoylation prediction tools, such as SUMOgo and JASSA, etc.

Reviewer 2 ·

Basic reporting

Clear and unambiguous, professional English used throughout.

Experimental design

Methods described with sufficient detail & information to replicate.

Validity of the findings

Conclusions are well stated, linked to original research question & limited to supporting results.

Reviewer 3 ·

Basic reporting

All my concerns are addressed and the quality of the paper has been improved. I have no further comments.

Experimental design

Good!

Validity of the findings

Good!

Additional comments

NA

Reviewer 4 ·

Basic reporting

Improved as per previous comments

Experimental design

Improved as per previous comments

Validity of the findings

Improved as per previous comments

---

## Round 0.3 · accepted · Accept

Your manuscript is now suitable for publication. Thanks for the contribution!